# Single-cell profiling reveals that dynamic lung immune responses distinguish protection from susceptibility to tuberculosis

Fergal J. Duffy[1], Maxwell L. Neal[1], Courtney R. Plumlee[1], Sara B. Cohen[1], Benjamin H. Gern[1,2], Alan H. Diercks[1], Michael Y. Gerner[3], Kevin B. Urdahl[1,2,3]*, John D. Aitchison[1,2]*

1 Center for Global Infectious Disease Research, Seattle Children's Research Institute, Seattle, Washington, United States of America, 2 University of Washington, Department of Pediatrics, Seattle, Washington, United States of America, 3 University of Washington, Department of Immunology, Seattle, Washington, United States of America

* Kevin.Urdahl@seattlechildrens.org (KBU); John.Aitchison@seattlechildrens.org (JDA)

## Abstract

The mechanisms that underlie protective immunity to Mycobacterium tuberculosis (Mtb) remain incompletely defined. To identify immune correlates associated with protection, we performed single-cell RNA sequencing of lung immune cells after aerosol Mtb infection of naïve mice and mice with contained Mtb infection (CoMtb), a model of naturally acquired resistance, across multiple time points, mouse strains, and Mtb strains. Protection was associated with distinct temporal patterns of immune activation, cell recruitment, and resolution. Early after challenge, CoMtb mice exhibited rapid, transient recruitment and activation of CD4$^+$T cells, macrophages, NK, and NKT cells, accompanied by short-lived bursts of type I and II interferon signaling, increased oxidative phosphorylation, and enhanced chemokine-mediated cell–cell communication. In contrast, primary infection elicited delayed but sustained interferon and neutrophil responses and higher bacterial burdens. These data indicate that protection involves dynamically coordinated immune pathways rather than the magnitude of any single response. Transcriptional features of CoMtb overlapped with those observed in nonhuman primates following intravenous BCG vaccination, including enrichment of activated tissue-resident CD4$^+$T cells and innate effector populations. Together, these findings support a model in which effective immunity to Mtb depends on the timing and coordination of immune activation, providing a framework for vaccine strategies that reproduce protective lung immune dynamics.

## Author summary

Mycobacterium tuberculosis (Mtb) is the most deadly individual infectious pathogen on earth, killing 1 million people per year. The sole existing vaccine, BCG,

**Data availability statement:** Data are available in the article itself and its supplementary materials. scRNAseq data are available from NCBI GEO and under the accessions: GSE266233 [B6], GSE266432 [B6 Sp140-/-], and GSE265843 [C3H]. Data include processed, annotated R Seurat analysis objects, and links to raw sequencing read fastq files freely available via NCBI SRA via BioProject IDs PRJNA1106428 [B6 scSCRNAseq], PRJNA1104528 [C3H scRNAseq] and PRJNA1107174 [B6 Sp140-/- scRNAseq].

**Funding:** This work was supported by the National Institutes of Health (NIH) contract 75N93019C00070 (K.B.U., M.Y.G.). The funders played no role in the study design, data collection, analysis, decision to publish or preparation of this manuscript.

**Competing interests:** The authors have declared that no competing interests exist.

is effective at preventing severe TB disease in children, but not at stopping the spread of the bacteria, therefore a more efficacious vaccine is needed. Recent work in animal models has shown promise at identifying immune interventions that can control TB. One example is the CoMtb model, where a high dose of Mtb bacteria are inoculated and contained in a mouse ear, which effectively reduces bacterial burdens in mice upon aerosol Mtb challenge. Here, we use single cell RNA sequencing of mouse lungs over an Mtb infection time course to identify CoMtb-induced protective factors in differently TB-susceptible mouse strains. Importantly, by looking at very early timepoints in infection, we were able to control for the effects of differing Mtb burdens in the lung. We show that CoMtb facilitates more rapid recruitment of multiple key anti-TB immune cell types to the lung, such as CD4 + T cells. Understanding how to elicit these protective responses in a vaccine suitable for human use could make a major impact in TB elimination

## Introduction

Developing a more efficacious TB vaccine would save millions of lives. While the existing BCG vaccine protects children against disseminated TB its efficacy in preventing infection and transmission in adults is limited. Critically, the immune responses necessary and sufficient to contain or control Mtb infection remain unclear. This lack of surrogate markers of effective TB immunity has made vaccine trials difficult, expensive, and slow.

Directly investigating the immune response to Mtb infection in the lung requires animal models, most commonly mice. Genetically identical mice infected with a supraphysiologic [1] conventional dose (50–100 CFU) of laboratory strain H37Rv Mtb invariably develop visible signs of disease coupled with a high burden of Mtb in the lung at 2–4 weeks post infection. We have previously shown that establishment of a contained Mtb infection in mice (CoMtb) in the ear draining lymph node also protects against subsequent Mtb aerosol challenge as well as against other bacterial infections [2,3]. This is consistent with epidemiological observations in human populations, such as nursing students, where prior asymptomatic Mtb infection provides protection against the development of symptomatic disease following further Mtb exposure [4,5].

Time-course profiling of lung immune responses *in-vivo* is the most direct way to determine the immune processes required for Mtb control, and considerable work has been done to characterize these responses. Previous work in the mouse model by ourselves and others has implicated alveolar macrophages as an early replicative niche for Mtb following infection [6,7] whereas pro-inflammatory recruited macrophages restrict Mtb growth [8,9]. Research has also examined the roles of type I interferon (IFNα and IFNβ) and type II interferon (IFNγ) in the lung. Elevated type I interferon responses have been implicated in disease progression [10,11], whereas type II interferon has been associated with disease control [12]. Recent advances in Mtb mouse model analysis have focused on identifying correlates of immunity using

either multiple inbred strains with varying susceptibility to Mtb [10,13,14] or genetically outbred mice infected with diverse Mtb strains [15]. These studies have linked increased proportions of lung resident CD4+ and CD8+T cells and B cells with Mtb control, whereas increased neutrophil proportions correlated with greater Mtb burdens in the lung.

Moving from correlative studies of factors linked to Mtb protection to mechanistic investigations has been challenging, particularly given the need to distinguish immune processes that merely reflect bacterial burden, from those that are directly associated with Mtb control. Recently, we showed that depletion of neutrophils in the C3HeB/FeJ mouse late in infection can disrupt necrotic lung pathology [16]. Here, building on this work, we explore single-cell transcriptomes of lung immune responses to Mtb infection over time and across multiple mouse and Mtb strains. We sought to determine whether immune correlates of protection could be identified independently of bacterial burden. Accordingly, we show that by 10 days after Mtb infection, when Mtb burden is equivalent across multiple mouse strains and between CoMtb and primary Mtb infection, multiple immune cell types and transcriptional states correlate with Mtb protection, independent of Mtb bacterial burden. These findings provide a foundation for distinguishing causal mechanisms of protection from downstream consequences of bacterial burden and could inform new candidates for surrogate outcome markers in vaccine trials as well as targets for host-directed Mtb therapies.

## Results

### Single-cell profiling of Mtb infected mouse lungs

To determine immune cell types and transcriptional responses associated with CoMtb-induced protection against Mtb infection, we performed single cell RNAseq profiling using the 10X Chromium platform. Mouse lungs were sampled at multiple time points after conventional dose (with a target of 50–100 CFU) aerosol Mtb infection across multiple combinations of Mtb strains and mouse genetic backgrounds, to compare differences in Mtb-resistant and susceptible host responses in tandem with differences in bacterial pathogenicity (Table 1). Two mouse strains were tested: the more Mtb-resistant C57/BL6 [17] and more Mtb-susceptible C3HeB/FeJ strain [18]. We also tested two Mtb strains: the standard laboratory strain H37Rv and, SA161, a hypoimmunogenic and hypervirulent clinical isolate [19]. Prior to scRNAseq profiling, lung cells were sorted after *in-vivo* IV antibody staining to exclude circulating blood cells present in the lung vasculature and retain only resident lung cells. After QC filtering, a total of 89 scRNAseq samples were obtained (S1 Table), with at least three mice for each combination of mouse strain, Mtb strain, timepoint and CoMtb status, however only data for Mtb SA161 was obtained for day 10 infections. The overall experimental design is shown in Fig 1A.

Lung Mtb CFU was measured for each mouse (Fig 1C). CoMtb mice showed significantly reduced Mtb burden compared with primary Mtb infection across all mouse/Mtb strain combinations at days 17 and 34 post infection. At day 10, however, lung Mtb burdens were similar across mouse strains and between CoMtb and primary infection, averaging

**Table 1. Mouse lung scRNAseq samples. n = 4 mouse lungs were sampled for each combination of timepoint, mouse strain, Mtb strain, and CoMtb status listed below.**

| Timepoints | Mouse strains | Mtb strains | CoMtb status |
|---|---|---|---|
| Pre-infection/Day 0 | C57/BL6 | H37Rv, SA161 | CoMtb, Primary Infection |
| | C3HeB/FeJ | H37Rv, SA161 | CoMtb, Primary Infection |
| Day 10 | C57/BL6 | SA161 | CoMtb, Primary Infection |
| | C57/BL6 Sp140-/- | SA161 | CoMtb, Primary Infection |
| | C3HeB/FeJ | SA161 | CoMtb, Primary Infection |
| Day 17 | C57/BL6 | H37Rv, SA161 | CoMtb, Primary Infection |
| | C3HeB/FeJ | H37Rv, SA161 | CoMtb, Primary Infection |
| Day 34 | C57/BL6 | H37Rv, SA161 | CoMtb, Primary Infection |
| | C3HeB/FeJ | H37Rv, SA161 | CoMtb, Primary Infection |



**Fig 1. Profiling lung immune responses to Mtb infection. A**. Illustrated study design showing mouse strains, Mtb strains, and sampling timepoints. Created in BioRender. Duffy, **F.** (2026) https://BioRender.com/jej2nqp. **B**. UMAP plot summarizing scRNAseq cell type annotation over all mouse samples at all timepoints (total n = 89) **C**. Mtb lung CFU post infection over time, stratified by mouse strain, Mtb strain, and CoMtb status. Individual dots represent CFU per mouse, and lines show group medians. Significance stars represent p-values determined by fitting negative-binomial (NB) linear models. **D,E**. Dotplot showing scRNAseq derived immune cell proportions (per thousand immune cells) in the lung over the course of infection, stratified by Mtb strain for C57/BL6 (D) and C3HeB/FeJ **(E)**. Purple dots indicate increased proportions in CoMtb vs. primary infection, vice-versa for green dots. Dot size is correlated with median cell type proportion in each mouse strain/Mtb strain/timepoint combination, using the more abundant of CoMtb vs Primary infection. Each mouse strain/Mtb strain/timepoint/CoMtb status combination included 3-4 mice.

approximately $10^4$ CFU. By day 34, C57/BL6 (B6) mice consistently exhibited lower bacterial burdens than C3HeB/FeJ (C3H) mice infected with the same Mtb strain and CoMtb status, consistent with the known increased susceptibility of the C3H genetic background to Mtb infection [20].

To quantify changes in immune cell proportion during infection, canonical immune cell type labels were assigned using a two-step process. Firstly, broad immune cell types (e.g., T cell, B cell) were assigned using celltypist [21] and a custom mouse immune cell reference library derived from scMCA [22] and Tabula Muris [23] mouse cell atlas resources (see Methods). This was followed by unsupervised clustering and manual annotation of finer-grained cell

subsets performed jointly for T cells, NK cells and ILCs, and separately for monocytes, macrophages and DCs. Within the resulting subclusters, canonical immune cell type labels were assigned by examining the expression of known cell lineage specific genes (**Figs 1B** and **S1B Fig**). Overall, we observed multiple distinct immune populations in the lung across all conditions (**S1B Fig).** These immune populations were then quantified per sample and standardized to cells-per-thousand total immune cells per sample.

## Mtb infection shifted lung immune cell compositions across mouse and Mtb strains

Post aerosol Mtb infection, we observed an increase in the proportion of many lung resident immune cell types between day 0 to day 17 across all samples (**Fig 1D** and **1E**). In particular, CD4+ and CD8+T cell subsets, monocyte-derived macrophages (MDMs), natural killer (NK) cells and neutrophils increased in proportion, while alveolar macrophage (AM) proportions declined (**Fig 1D** and **1E**). AM proportions continued to decline between day 17 to day 34, however other cell types exhibited diverging patterns, primarily associated with the presence of CoMtb. Overall, changes in cell composition were similar between H37Rv and SA161 Mtb infection, despite variation in aerosol Mtb dose measured in inoculation control mice for H37Rv (~141 CFU) and SA161 (~14 CFU). However, stark differences were apparent between mouse strains pre-infection, and early post-infection (days 0 and 10), with C3H mice showing higher proportions of AMs and lower proportions of all other immune cells compared to B6 mice (**Figs 1E** and **S2**). These reduced lung immune cell proportions at early timepoints in C3H mice are consistent with the known increased Mtb susceptibility of this strain to Mtb infection, and the higher lung Mtb burdens observed at day 34 (**Fig 1C**).

## CoMtb was associated with faster recruitment of many immune subsets to the lung

Prior to aerosol infection (day 0), CoMtb mice with both C3H and B6 backgrounds had higher proportions of activated CD4+T cells in the lung (**S2 Fig**). Following infection, CoMtb was also associated with more rapid increases in activated CD4+ and CD8+T cells, MDMs, NK cells and neutrophils than primary infection in both mouse strains (**S3** and **S4A Figs**). Although the abundance of these cell types generally tracked Mtb burden, in CoMtb C57BL/6 mice, the proportions of NK cells, MDMs and neutrophils peaked by day 10 post infection (**S4A Fig**), when lung Mtb CFUs were still low. This suggests that CoMtb may promote recruitment of these cell types independent of Mtb burden. As an independent comparison, IV-cell counts from flow cytometry of separate mouse infection experiments with matching experimental designs were used to quantify cell types corresponding to those identified by scRNAseq (**S4B** and **S4C Fig**). Cell populations were defined using a previously designed surface marker panel and gating strategy [16]. In these flow panels, consistent with scRNA-seq results, activated CD4+ (CD44+) T cells, MDMs, and neutrophils were increased at day 10 and day 17 in CoMtb, particularly in B6 mice. NK counts were not determined in C3H mice as they lack the NK1.1 marker used in this panel. In contrast to the reduction in AM proportions observed over time in scRNAseq data, flow cytometry counts indicated that absolute AM numbers were relatively stable or slightly increased during infection. Thus, the apparent decline in AM proportion observed in scRNAseq samples (**Fig 1D** and **1E**) likely reflects changes in other cell populations, rather than a loss of AMs in the lung.

## CoMtb was associated with reprogramming of immune cells during Mtb infection

In addition to changes in cell abundance, we examined whether CoMtb altered transcriptional programs within immune cell subsets. A pseudobulk strategy was applied, aggregating read counts per-sample and cell type to identify changes in transcriptional programming within cell types comparing post-infection time points to pre-infection. In CoMtb B6 and C3H mice, hundreds to thousands of differentially expressed genes (DEGs) were identified in each immune subset comparing early timepoints (day 10 and day 17) to pre-infection (**S5A Fig**), whereas relatively few DEGs were observed at day 34. In CoMtb mice, SA161 infection produced a rapid and transient increase in DEG numbers peaking at day 10 and declining by day 17, whereas gene expression changes in H37Rv-infected peaked at day 17 and declined by day 34. Overall, B6

mice exhibited more DEGs than C3H mice, although this could reflect limited statistical power due to the low numbers of non-AM immune cells in C3H mice at day 0. Consistent with this, in primary Mtb infection, DEG counts were substantially lower at early time points in both mouse strains, except in the AM subset, and increased steadily to peak at day 34. Thus, CoMtb was associated with more rapid and transient transcriptional responses to infection across most immune cell types, while SA161 induced faster and broader responses than H37Rv.

To interpret DEGs functionally, we performed gene set enrichment analysis (GSEA) using transcriptionally coherent immune gene sets from mSigDB Hallmark collection [25]. Mtb responsive gene sets were then selected as those that met the threshold of GSEA |NES| > 2 for at least one cell type comparing day 0 to later timepoints post SA161 infection (**Fig 2A**) and post H37Rv infection (**S5B Fig**). We identified 12 transcriptionally coherent pathways differentially expressed across multiple cell types and conditions: oxidative phosphorylation (OXPHOS), *Myc* targets V1 and V2, DNA repair, *E2F* targets, *IL6*/*JAK*/*STAT3* signaling, allograft rejection (involving MHC-I signaling to CD8), the inflammatory response, *TNF* signalling via *NFkB* and the type I (alpha) and II (gamma) interferon responses.

At day 10 post SA161 infection, CoMtb B6 and C3H mice showed strong decreases in transcripts associated with OXPHOS, *Myc* targets and DNA repair pathways across almost all cell types, with the most pronounced effects in AMs (**Fig 2A**). Few significant changes in pathway responses were detected in C3H primary infection at day 10. In B6 mice at day 10, following primary infection, reductions in OXPHOS pathway transcripts were observed in most T cell and lymphoid subsets as well as MDMs, but not in AMs, cDCs or pDCs. By contrast, CoMtb mice at the same time point exhibited strong upregulation of transcripts in inflammatory response, *TNF,* and interferon response pathways for B cells, neutrophils, cDCs, MDMs, and AMs, while effects in CD4+ and CD8 + T cell subsets were weak or not significant. Primary infected B6 mice at day 10 also showed some increase in pro-inflammatory processes particularly in cDCs, neutrophils and inflammatory AMs.

By days 17 and 34 transcriptional changes relative to day 0 were broadly similar across all combinations of mouse and Mtb strain combinations. In primary infection both B6 and C3H mice showed very strong increases in type I and type II interferon pathway transcripts across almost all cell types, with neutrophils also upregulating in *TNF* signaling genes, and both neutrophils and monocyte-derived cells upregulating inflammatory pathway genes. Primary infection was further associated with increased OXPHOS pathway and *Myc* target transcripts at day 34 affecting nearly all cell types in B6 and specifically naïve T cells, Tregs and MDMs and AM subsets in C3H. In contrast, OXPHOS and *Myc* targets were decreased or unchanged in all immune subsets in CoMtb. Certain responses at days 17 and 34 were highly attenuated in B6 CoMtb mice compared with primary infection. In particular, increases in type I and II interferon responses relative to day 0 were restricted to AMs, cDCs, neutrophils and naïve CD4 T cells. C3H CoMtb mice showed a broadly similar pattern to C3H primary infection in terms of interferon responses, although the very small numbers of non-AM immune cells present in the C3H lung at day 0 (**S2 Fig**) may have limited statistical power relative to B6. Overall, this analysis showed primary SA161 infection was associated with sustained increases in type I interferon signaling in both C3H and B6 mice, but this response was attenuated in the more Mtb-resistant B6 + CoMtb group, where type I IFN signaling peaked at day 10 post aerosol challenge.

Along with changes in transcriptional responses measured relative to the day of infection (**Figs 2A** and **S5**B), we assessed changes associated specifically with CoMtb. **Fig 2B** shows GSEA NES scores for Mtb-responsive gene sets in CoMtb vs uninfected mice prior to aerosol infection (day 0). Transcriptional differences were most apparent in B6 mice, where CoMtb was associated with robust upregulation of genes linked to OXPHOS, *Myc* targets, DNA repair and interferon responses. OXPHOS and *Myc* target responses were highly correlated and increased in most B6/CoMtb cell types, with the notable exception of neutrophils, ki67 + T cells and pDCs. In C3H mice, CoMtb was associated with increased OXPHOS transcripts in naïve CD4 + T cells and AMs. Both type I and type II interferon responses were elevated in CoMtb relative to primary infection, but the cell-type specificity and timing of these responses differed. In B6 mice, interferon gamma responses were increased in nearly all cell subsets except ki67 + replicating AMs, while type I interferon responses

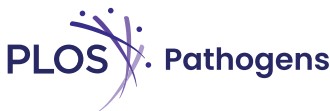

**Fig 2. CoMtb induced changes in gene expression pathways. A.** Heatmap indicating changes in GSEA normalized enrichment score (NES) for Mtb-responsive gene sets calculated from cell type-specific pseudobulk scRNA-seq fold changes in SA161 infected mice relative to day 0 (3-4 samples per group). Dot sizes increase with decreasing FDR, and color indicates NES. Significant pathways (FDR<0.05) are shown as solid dots. Pathways with any |NES|>2 are shown. **B**. Heatmap indicating in NES for CoMtb vs uninfected mice at day 0.

were increased in most cell types except MDMs, pDCs, gdTs and ki67+T cells. In C3H, interferon gamma responses were observed in in AMs, NKTs and both activated and naïve CD4 cells, but type I interferon responses were not significantly elevated in any subset. Thus, although CoMtb mice showed increased inflammatory signaling at day 10 compared with pre-infection, they began with a higher inflammatory 'baseline' relative to naïve mice, particularly for interferon gamma response genes.

## CoMtb induced protection-associated cell types independent of Mtb burden

We observed that CoMtb was associated with improved Mtb protection, more rapid recruitment of specific immune subsets to the lung, and broad rapid and transient, as opposed to sustained, changes in transcriptional programs within immune subsets post infection. However, linking these changes directly with improved clearance or containment of Mtb infection is complicated by two main factors. First, many of these CoMtb-associated changes in cell type abundance were also correlated with Mtb burden (S3 Fig). Second, interpretation is confounded by the large baseline differences in lung immune populations (Fig 1C) between more susceptible C3H and less susceptible B6 mice, making it unclear clear which immune cell types differing between the mouse strains are mechanistically involved in protection.

To more directly identify cell types and responses associated with protection independent of bacterial load, we focused on day 10 post-infection when lung Mtb burdens were equivalent across mouse strains and between primary infection and CoMtb (Fig 1C), allowing differences in immune cell proportions to be interpreted independent of bacterial loads. To limit confounding effects of complex genetic differences between C3H mice and BG mice, we examined B6 mice lacking *Sp140,* a type I interferon regulator within the *Sst1 (Super susceptibility to tuberculosis 1)* locus, whose loss recapitulates the increased susceptible observed in C3H mice [10]. B6 Sp140-/- mice were added to the day 10 analysis as a susceptible comparator alongside B6 wild type (WT) and C3H mice, allowing us to control for genetic background while comparing divergent infection outcomes in B6 WT mice with or without CoMtb. Consistent with this design, B6 *Sp140-/-* mice had similar day 10 lung CFU burdens as B6 WT mice, and these were not significantly affected by CoMtb (Fig 3A). Testing for changes in immune subset proportional abundance changes at day 10 revealed eight cell types that were significantly increased in CoMtb vs primary SA161 infection in both B6 WT and B6 *Sp140-/-* (Fig 3B), indicating that they were positively associated with protection from Mtb independent of bacterial load across two otherwise similar genotypes with differing susceptibility. These included multiple T cell subsets, along with MDMs, neutrophils, and NK cells. Notably, two of these cell types (NKT cells and activated CD4+T cells) were also significantly reduced in B6 *Sp140-/-* vs B6 WT – that is, they were both enriched in the protective CoMtb state and depleted in the more susceptible genotype.

As an independent comparison, flow-cytometry determined IV- cell counts were examined for equivalent cell types at day 10 post SA161 infection (Fig 3C) comparing CoMtb to primary infection. MDM and neutrophil counts were strongly increased in both B6 WT and B6 *Sp140-/-* CoMtb mice, while activated CD4+ and CD8+cells were only significantly increased in B6 WT CoMtb mice. This discrepancy may reflect differences in how T cell subsets are defined by RNA expression versus surface protein markers, suggesting that the precise transcriptional activation state CoMtb-induced T cells is important for protection. To investigate this further, we investigated transcriptional changes within these cell types at day 10 post infection.

## CoMtb induced interferon and inflammatory pathways independent of Mtb burden

Transcriptional changes at day 10 were identified by using the pseudobulk approach as above, comparing CoMtb vs primary infection in B6 WT and B6 *Sp140-/-,* and B6 WT vs B6 *Sp140-/-* with or without CoMtb. B6 *Sp140-/-* showed interferon pathway responses similar to B6 at day 10 post infection (Fig 4A), contrasting with prior reports implicating *Sp140* in driving strong interferon responses 28 days after Mtb challenge [10,26]. Additionally, B6 WT mice showed marked decreases in the expression of genes associated with OXPHOS, *Myc* target and DNA repair pathways relative to B6 *Sp140-/-,* suggesting that the absence of *Sp140* is associated with increased proliferative or metabolic transcriptional responses early after Mtb challenge.

CoMtb associated pathway changes at day 10 were highly consistent between B6 WT and B6 Sp140-/- (Fig 4B) with sharp increases in type I and II interferon response pathway expression across most cell types, most prominently in B cells, activated CD4+T cells, AM, MDM and cDC subsets. This is consistent with recent work showing that CoMtb-mediated protection is elicited in part by enhancing the response of monocyte-derived cells to IFNγ [27]. The sole



**Fig 3. Immune cell types associated with CoMtb-mediated protection. A.** Mtb lung CFUs for B6 WT and B6 Sp140-/- mice at day 10. **B.** Box- and dot-plots showing scRNAseq quantified cell types significantly increased in CoMtb vs primary infection in B6 WT and B6 Sp140-/- mice 10 days after aerosol challenge with Mtb SA161. Each dot represents one mouse. **C** Box and dotplots showing total IV- flow cytometry determined cell counts for the subsets of cell types shown in **B**, derived from independent experiments. P-values indicated are from NB linear model fits.

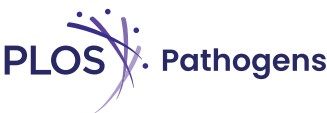

**Fig 4. CoMtb induced changes in gene expression pathways.** Heatmaps indicating changes in GSEA normalized enrichment score (NES) for Mtb-responsive pathways calculated from cell type specific pseudobulk scRNAseq gene fold changes at day 10 after aerosol Mtb infection. **A.** Comparison of B6 WT and B6 Sp140-/- for both CoMtb and primary infection. **B.** Comparison of CoMtb and Primary infection for both B6 and Sp140-/- mice.

exception was activated CD8 + T cells, which showed increased type I interferon responses in B6 *Sp140-/-,* but not in B6 WT mice. CoMtb was also associated with upregulation of other pro-inflammatory and immune signaling gene pathways (*TNF* signaling, inflammatory signaling, and *IL6* responses) in specific cell types - primarily B cells, neutrophils and AMs and, to a lesser extent, MDMs. These inflammatory responses were accompanied by decreased expression of OXPHOS and *Myc* target pathway transcripts in AMs and B cells.

The pronounced transcriptional differences between B6 WT and B6 *Sp140-/-* and between primary infection and CoMtb suggest that the increased susceptibility of B6 *Sp140-/-* mice and the protection conferred by CoMtb were mediated by distinct host immune pathways.

To complement these pseudobulk analyses, we performed unsupervised clustering within labelled scRNAseq cell types that were increased in CoMtb at day 10 (**Figs S1**B and **3B**), using only day 10 samples from B6 and B6 Sp140-/- mice. By separating cell types prior to reclustering, this analysis directly examined within cell-type transcriptional heterogeneity by focusing only on genes robustly expressed within each population. This approach revealed striking CoMtb-associated differences in MDMs, with minimal differences in B6 WT and B6 Sp140-/- MDMs during primary infection (**Fig 5A**). In MDMs, primary Mtb infection was characterized by a single homogeneous MDM subcluster (subcluster 2) expressing macrophage-associated markers such as *Mrc1* as well as as *Cd81*, *Cd63 Timp2* and *C1qa* (**Fig 5B**), which were not expressed, or only weakly expressed in other MDM subclusters. In contrast, CoMtb-enriched subclusters (0, 1, 3) all exhibited increased expression of *Isg15*, *Ly6c2* and *Zpb1*. Subcluster 0 was further distinguished by high levels of *Nos2* and *Inhba*; while subcluster 1 specifically expressed *Ifitm6* and *Sell*. Consistent with the elevated *Isg15* expression, interferon stimulated gene expression was increased across all CoMtb-associated MDM subclusters (**Fig 5C**).

For other cell types, particularly lymphocytes, subcluster structure was strongly influenced by both *Sp140-/-* status and CoMtb status (**S6 Fig**). Differentially expressed genes associated with *Sp140-/-* status overlapped extensively among lymphocyte subsets (**S2 Table**), with 107 Sp140-/- associated DEGs shared across all lymphocytes. These shared genes were enriched for ribosomal and transcription-associated functions, and 69 of them overlapped with a single specific gene module: the MSigDB C4 cancer gene neighborhood MORF_TPT1 module. This module comprises genes transcriptionally correlated with *Tpt1* (Tumor Protein, Translationally-Controlled 1), a ubiquitously expressed, evolutionarily conserved multifunctional protein, associated with growth, proliferation, and anti-apoptotic signaling, and poor cancer outcomes [28]. **Fig 5D-K** shows expression scores for the MORF_TPT1 module, showing that Sp140-/- mice exhibit increased expression of MORF_TPT1 genes across all lymphocyte populations (**Fig 5D-I**) regardless of CoMtb status. This effect was notably attenuated in myeloid MDM and monocyte populations (**Fig 5J** and **5K**). Together, these findings suggest that CoMtb mediated protection may arise from early reprogramming of lung-recruited MDMs, whereas the increased susceptibility of Sp140-/- may be linked to *Tpt1*-associated proliferative signaling in lymphocytes.

## CellChat revealed increased immune effector signaling associated with protection

Along with changes in abundance and within-cell transcriptional programs, we hypothesized that CoMtb might mediate protection through altered intercellular signaling. To investigate this, we applied CellChat [29] to estimate directed cell-cell communication between cell types increased in abundance by CoMtb at day 10 (**Fig 3B**). CellChat infers cell-cell communication between a 'source' cell type expressing a ligand and 'target' cell type expressing the corresponding cell surface receptor. Communication probabilities are calculated using a mass-action model with receptor and ligand abundance estimated based on per-cell gene expression. Communication networks were estimated for both B6 and B6 *Sp140-/-* mice 10 days post SA161 infection comparing CoMtb and primary infection and included both autocrine and paracrine signaling.

Differences in cell-cell communication between CoMtb and primary infection are shown in **Fig 6**. Overall, multiple signaling pathways associated with immune effector functions were altered in CoMtb compared with primary infection, and CoMtb-induced changes were highly consistent in both B6 WT (**Fig 6A**) and B6 *Sp140-/-* (**Fig 6B**) mice. The interaction between *App* (Amyloid precursor protein) – *Cd74,* previously shown to suppress macrophage phagocytosis [30], was reduced in CoMtb, specifically in communication between MDMs and neutrophils to MDMs. CoMtb was also associated with reduced communication from *Ptprc* (*Cd45*), expressed by all protection associated cell types, to the *Mrc1* mannose receptor on MDMs. In addition, *Cxcl2-Cxcr2* communication from MDMs and neutrophils to neutrophils was diminished in CoMtb. Because Cxcl2-Cxcr2 plays a role in neutrophil recruitment and swarming [31], this may reflect reduced neutrophil



**Fig 5. Subclustering reveals CoMtb and Sp140-/- associated changes within cell types. A.** UMAP plots showing scRNAseq identified MDM subclusters in B6 WT and B6 Sp140-/- mice with and without CoMtb (n = 3-4 per group) at day 10 post infection. **B.** Expression dotplots of marker genes that significantly discriminate MDM subclusters. **C**. Violin plots showing per-cell interferon gene module score distributions for each MDM subcluster. **D-K**. Violin plots showing per-cell module expression scores for the MSigDB TPT1 cancer gene network module (MORF_TPT1) for each day 10 CoMtb-associated cell type.

activity associated with CoMtb at sites of infection. Finally, CoMtb was associated with reduced MHC-II communication from MDMs, suggesting reduced antigen presentation from MDMs to CD4 T cells at this timepoint.

In contrast, CoMtb was associated with increased pro-inflammatory CCL chemokine signaling (i.e., *Ccl5 − Ccr5* and *Ccl4 − Ccr5*), which promotes T cell recruitment [32,33], These signals were strongest from activated CD8+T cells, NK and NKT cells to activated CD8+T cells, MDMs and NK cells. CoMtb also enhanced *Cd80 − Cd274* (*PD-L1*) signaling



**Fig 6. CoMtb-induced changes in cell-cell communication.** Dotplots showing CoMtb-associated changes in interaction strength, estimated by CellChat from scRNAseq-quantified cell types, for **A**. B6 WT and **B**. B6 Sp140-/- mice 10 days post SA161 infection. Receptor-ligand interaction pairs are shown on the y-axis and grouped into pathways on the right of the plot. "Sender" cell types (expressing ligands) are indicated above each column and "Target" cell types (expression receptors) are shown on the x-axis. Dot size indicates the maximum interaction strength in either CoMtb or primary infection, and dot color indicates CoMtb change: red = increased in CoMtb, blue = decreased in CoMtb.

from MDMs and neutrophils to all other protection associated cell types. The *Cd80 – PD-L1* complex promotes optimal T cell activation by preventing the T cell inhibitory *PD-1*/*PD-L1* interaction [34]. In addition, CoMtb was associated with increased *Icam1 (Cd54) - Itgal (Cd11a)* signaling among protection-associated cell types, an interaction associated with

T-cell migration [35]. Whereas MHC-II signaling from MDMs to CD4+T cells was reduced, MHC-I signaling from multiple immune cell types to CD8+T cells was increased n CoMtb. More complicated patterns were seen for *Itgb2-Cd226* communication. Engagement of T cell surface *Cd226* promotes the cytotoxic activity of CD8+T cells [36]. In B6 WT, CoMtb was associated with increased *Itgb2-Cd226* communication from multiple cell types to CD8+T cells but reduced signaling to CD4+T cells. In contrast, *Itgb2-Cd226* signaling both CD4+ and CD8+T cells was increased in B6 *Sp140-/-* mice.

Taken together, CoMtb was associated with increased signaling through pathways that promote immune activation and migration, while reducing communication linked to immune inhibition and neutrophil recruitment. Overall, CoMtb promotes more effective Mtb immunity characterized by recruitment of multiple immune cell types to the lung, reprogrammed gene expression programs, and coordinated cell-cell crosstalk facilitating T cell migration and activation.

## Discussion and conclusions

This work provides a detailed picture of the lung immune response to Mtb infection in mice over time, contrasting concomitant protection (CoMtb) with Mtb susceptibility related to genetic background (C57BL/6 vs C3HeB/HeJ and C57BL/6 Sp140-/-). Overall, Mtb infection was associated with a shift in lung immune cell proportions, from one dominated by AMs to a full complement of lung-resident myeloid (e.g. MDMs, Neutrophils) and lymphoid (B, T and NK cells) populations. This was accompanied by within-cell increases in type I interferon gene expression across diverse immune cell types. While this response was broadly observed in all mice after infection, the timing and magnitude of gene expression changes across cell populations varied markedly depending on mouse genetic background and whether infection was primary or in the presence of CoMtb.

To help distinguish responses that directly control Mtb burden from those that simply respond to Mtb burden, or reflect mouse genotype-specific differences, we focused on an early timepoint (10 days post infection) when lung bacterial burdens had not yet diverged between groups. At this time point, CoMtb was associated with increased levels of activated T cells, MDMs, neutrophils, NK and NKT cells in both B6 WT and B6 Sp140-/- mice. These cell types exhibited elevated expression of type I and II interferon response genes, indicating that CoMtb significantly increased the recruitment and activation of multiple interferon-responsive immune populations independently of lung Mtb burden. Notably, CoMtb mice also had higher levels of lung-resident activated CD4+T cells both prior to and following aerosol challenge, suggesting that some may be Mtb-specific and contribute to an accelerated adaptive immune response relative to primary Mtb infection. In contrast, primary infections showed delayed induction of type I and II interferon responses but more prolonged activation, with sustained interferon responses persisting through day 34, whereas CoMtb responses were more rapid and transient.

Intravenous BCG (iv-BCG) has been shown to be highly protective against Mtb infection in non-human primates (NHPs) [37]. Importantly, CoMtb-associated immune features observed in the mouse lung parallel those reported for iv-BCG mediated protection in NHP lungs. Specifically, Peters et al. [38] showed that high-dose iv-BCG vaccination elicited increased airway T cells and recruited macrophages, and enhanced interferon signaling following stimulation with mycobacterial antigen compared with unvaccinated NHPs. Additionally, Simonson et al. demonstrated that the presence of sufficient levels of lung resident activated CD4+T cells, NK cells, and innate T cells are necessary for iv-BCG mediated protection in the NHP TB model [39].

A negative correlation between type I interferon and TB disease outcomes has been extensively explored in mouse models [10,11,40] and type I interferon has also been associated with the development of active TB in humans [41–43]. In contrast, type II interferon produced by CD4+T cells is necessary for Mtb control [12,44]. In our dataset expression of type I and type II IFN response genes tended to be correlated. This may reflect crosstalk in downstream responses. Therefore, examining interferon-responsive transcripts alone may not reliably discriminate the effects of type I vs type II interferon. However, there were instances in our data in which the type II interferon signatures were stronger than type I responses. Prior to aerosol infection, significant type II interferon responses seen in AMs, NKTs, and activated CD4+T cells from CoMtb C3HeB/FeJ mice relative to uninfected controls, while no cell types showed a significant type I interferon response.

In C57BL/6 mice pre-infection, most cell types exhibited both type I and type II interferon responses in CoMtb compared to uninfected mice but type II responses were broadly stronger. This suggests that CoMtb may preferentially enhance type II interferon signatures prior to infection. This observation is supported by recent work [27] showing that CoMtb was associated with increasing responsiveness to IFNγ signaling in monocyte derived cells, and counteracts the reduction in MHC-II expression in monocyte derived cells associated with exposure to type I interferons.

The increased susceptibility of Sp140-/- mice has been attributed to their reduced ability to repress type I interferon signaling, as Ji et al [10] showed that Mtb-infected Sp140-/- mice have significantly elevated type I interferon expression by 28 days post infection. Additionally, Kotov et al [11] showed that depletion of type I interferon-producing pDCs 25 days post infection in Sp140-/- mice reduced lung Mtb burdens. However, we did not observe elevated interferon responses in Sp140-/- mice relative to B6 at day 10 post infection, suggesting that differences in type I interferon responses between wild type and Sp140-/- mice may only become apparent several weeks after aerosol Mtb infection. Taken together, these findings suggest that transient increases in type I interferon responses very early in infection may be protective or neutral, whereas sustained type I interferon signaling at later stages contributes to increased Mtb tissue burden.

Our observation that interferon responses did not differ between Sp140-/- and WT B6 mice at day 10 contrasts with recent work [26] describing the role of *Sp140* as a negative regulator of IFNβ, with B6 *Sp140*-/- mice exhibiting significantly increased type I IFN responses 25–28 days post infection [10,11] – a phenotype alleviated by knockout of the type I interferon receptor (*IFNAR*-/-). It is likely that day 10 post infection is too early for the negative regulatory role of *Sp140* on IFNβ to become apparent. Nonetheless, we observed significant upregulation of *Tpt1*-related transcripts and the associated MORF_TPT1 module in *Sp140*-/- mice. *Tpt1* (aka fortilin) is an essential [45], ubiquitously expressed, highly conserved gene [46], that functions in multiple signaling pathways including DNA repair, p53 and anti-apoptotic signaling and is representative of proliferative programs linked to adverse cancer outcomes. This suggests that *Sp140* regulates cellular responses beyond IFNβ signaling, although which additional pathways underlie the observed differences in Mtb infection outcomes remain unclear.

In addition to interferon responses, CoMtb was associated with increased expression of OXPHOS, *Myc* targets and DNA repair pathways prior to aerosol Mtb infection, followed by reduced activity in these pathways at post-infection time-points compared to primary infection. The OXPHOS pathway is a key immunometabolic pathway, linked to cellular activation state [47]. OXPHOS is reduced in stimulated M1 macrophages [48] and DCs [49], and naïve T cells rely on OXPHOS [50], and switch to glycolysis upon activation [51]. Thus, prior to aerosol infection, in CoMtb mice, higher levels of OXPHOS activation may contribute to naïve-immune cell recruitment into the lung, consistent with the increased non-AM lung immune cell proportions seen in S2 Fig. Subsequent reductions in OXPHOS with CoMtb during infection may reflect more rapid immune maturation to M1 macrophages and activated T cells.

Two cell types were both increased in CoMtb and decreased in susceptible *Sp140*-/- at day 10: activated CD4 + T cells and NKT cells. Previous work has shown that NKT cells activated through the CD1d ligand α-galactosylceramide, but not unactivated NKTs, reduce mouse lung Mtb burden after aerosol infection [52]. Thus, CoMtb may provide the activation stimulus needed for effective NKT-mediated Mtb killing. The essential protective role of activated CD4 + T cells in murine TB, in the form of Mtb-specific IFNγ producing CD4 + T cells in response to infection has long been recognized [12,44,53]. However, vaccination strategies that expand circulating populations of Mtb-specific IFNγ producing T cells have failed to increase protection against TB in human trials [54,55]. This suggests that lung tissue residency of Mtb-specific CD4 + T cells may be a key determinant of protection.

CoMtb was also associated with recruitment of a wide number of activated innate immune cells to the lung. Identifying the minimal set of these activated cell types required for improved anti-Mtb responses may aid rational vaccine design strategies. Neutrophils, for instance, have been associated with negative outcomes in Mtb infection, due to interferon-associated hyperrecruitment leading to tissue damage [11,16]. However, we observed that neutrophil proportions were increased in CoMtb B6 and C3H mice at day 10, although by day 34 post infection neutrophils were higher in primary

infection for both B6 and C3H mice, with the very highest proportion being observed in the most susceptible primary infected C3H group. Neutrophils proportions were strongly correlated with Mtb burden across mouse strains, except in CoMtb B6 mice. Like interferon response dynamics, transient early neutrophil recruitment was not associated with increased Mtb burden, whereas sustained late recruitment was correlated with worse outcomes.

An important question is whether the cell types and responses associated with reduced bacterial burden in our study are also associated with protection in human or non-human primate models. Liu et al [56] identified blood transcriptional correlates of protection following efficacious IV-BCG vaccination [37] in rhesus macaques. Two days after vaccination, genes associated with type I interferon, neutrophil recruitment and monocyte activation were upregulated, followed by activation of CD4 + T cell and NK cell genes 2–4 weeks post vaccination. These patterns of expression closely mirror those observed here in CoMtb mice, where an early, transient wave of type I interferon signaling, and neutrophil recruitment was followed by increased induction of activated CD4 + T cells.

In conclusion, CoMtb reshapes the immune response to infection through rapid recruitment of activated T cells, NK and NKTs, MDMs and neutrophils accompanied by transient type I and II interferon expression. These changes parallel protection-associated responses elicited by iv-BCG vaccination in NHPs. In contrast, delayed immune cell recruitment to the lungs in primary infection led to sustained high levels of both type I interferon signaling and neutrophil recruitment later in infection. CoMtb-induced immunity was also characterized by altered cell-cell communication networks reduced immunosuppressive interactions and enhanced chemokine driven immune activation and recruitment. Together, these findings suggests that, in addition to the presence of Mtb-specific IFNγ producing CD4 + T cells, effective TB-immunity requires a lung immune milieu poised to rapidly respond and recruit activated innate cells from circulation.

## Methods

### Ethics statement

Experiments were performed with the approval of, and in compliance with, Seattle Children's Research Institute's Animal Care and Use Committee.

### Mice

C57BL/6 and C3HeB/FeJ mice were purchased from Jackson Laboratories (Bar Harbor, ME). All mice were housed in individually ventilated cages in specific pathogen-free conditions (maximum 5 mice/cage) within rooms with negative pressure ventilation and air filtering at Seattle Children's Research Institute. Animals were monitored under care of full-time staff, given free access to food and water and maintained under 12-hour light and dark cycles, with temperature controlled between 22–25 degrees Celsius. All possessed normal health and immune status. None had previous treatments, procedures, nor invasive testing prior to the initiation of our studies.

### Mtb Aerosol infections

Infections were performed as described previously with Mtb SA161 and H37Rv [57]. To perform CD aerosol infections, mice were placed in a Glas-Col aerosol infection chamber, and 50–100 CFU were deposited into their lungs. To confirm the infectious inoculum, two mice per infection were euthanized on the same day of infection, then their lungs homogenized and plated onto 7H10 or 7H11 plates for determination of CFU. A total of three independent infections were performed: (1) H37Rv for all mice at days 0, 17 and 34. (2) SA161 for all mice at days 0, 17 and 34 and (3) SA161 for all mice at day 10.

### Bacterial burden determination

Mouse lungs were individually homogenized in an M tube (Miltenyi) containing BS + 0.05% Tween-80. The resulting homogenates were diluted and plated onto 7H10 plates. Plates were incubated at 37 degrees Celsius for a minimum of 21 days before CFU enumeration.

### Concomitant Mtb model (CoMtb)

The CoMtb model was established as described previously [3]. Briefly, mice were first anesthetized by intraperitoneal injection of 400 ul of ketamine (4.5 mg/ml) and xylazine (0.5 mg/ml) diluted in PBS. Mice were placed in a lateral recumbent position, and the ear pinna was flattened with forceps and pinned onto an elevated dissection board using a 22 G needle. H37Rv Mtb grown to an OD between 0.2-0.5 over a 48-hour period was diluted to 106 CFU/ml in PBS, and 10 ul (104 CFU) was administered into the dermis of the ear using a 26s G Hamilton syringe. Mice were then rested for 6–8 weeks prior to subsequent aerosol challenge.

### Generation of single-cell suspensions

Prior to Mtb infection and at days 10, 17 and 34 post-Mtb infection, C57BL/6 and C3H/FeJ mice were anesthetized with isoflurane and administered 1 ug anti-CD45.2 antibody intravenously, along with C57BL/6 Sp140-/- mice at day 10 post Mtb infection only. After 5–10 minutes of in vivo incubation, mice were euthanized by CO2 asphyxiation. Mouse lungs were excised and lightly homogenized in HEPES buffer containing Liberase Blendzyme 3 (70 µg/ml; Roche) and DNaseI (30 µg/ml; Sigma-Aldrich) using a gentleMacs dissociator (Miltenyi Biotec). The lungs were then incubated for 30 min at 37°C and then further homogenized a second time with the gentleMacs. The homogenates were filtered through a 70 µm cell strainer, pelleted for RBC lysis with RBC lysing buffer (Thermo), and resuspended in FACS buffer (PBS containing 2.5% FBS and 0.1% NaN3), to produce a single-cell suspension.

### Single cell RNA sequencing

After generation of single cell suspensions, as described above, cells were resuspended in 200 µl MACS buffer (PBS containing 2.5% FBS plus 1 mM EDTA), filtered through a 70 µm filter, and run on a FACS AriaII (BD) sorter. To collect parenchymal cells for single-cell RNA sequencing, alveolar macrophages (AM, SiglecF+CD11c+) were sorted separately into one collection tube to account for autofluorescence in the IV label channel, and all other IV-negative cells were sorted into another collection tube. After sorting, the two populations were combined and counted on a hemocytometer. After one round of washing with ice-cold DPBS, cells were resuspended to 1000 cells/µl in DPBS, and 8000 cells were input into the 10X Genomics pipeline following the manufacturer's recommendations. After the generation of cDNA following the manufacturer's protocol, samples were centrifuged through two sequential rounds of 0.2 µm SpinX (Costar) columns to sterilize the sample for removal from the BSL3 facility and subsequent library generation. Libraries were submitted to Psomagen (Rockville, MD) for NovaSeq sequencing, with 300M reads per sample.

### Alignment and intial processing of single cell RNA sequencing data

10X Chromium 3' derived single-cell RNAseq sequence reads were aligned to the 10X Genomics pre-built mouse reference genome mm10–2020-A, assigned to individual cells by barcode, and UMI summarized using the 10X Cell Ranger 7.1.0 software package.

The Seurat R package was used for initial QC filtering and integration. First, a filtering step was applied across all samples, requiring all passing cells to have UMIs mapped to at least 500 distinct genes, and fewer than 5% of UMIs mapped to mitochondrial genes. Genes detected in fewer than 3 cells per mouse were excluded from further analysis. The Seurat integration pipeline [58] was then applied to correct for batch effects and align cells across conditions including all combinations of mouse strain, Mtb strain, time post challenge, and CoMtb status.

### Downstream single cell RNA sequencing analysis

Initial cell type assignment was performed using the CellTypist python package [21], As CellTypist does not have an available cell type model suitable for mouse lung or mouse immune cells, we created a de novo mouse lung immune cell type model using two published mouse cell atlases, namely the Tabula Muris [23] and scMCA [22] resources. Cell

type labels were harmonized between both sources (e.g., macrophage -> Macrophage) and both datasets were fil-tered to retain immune and lung-associated cell types, excluding cells specific to other organs. The CellTypist python package was then used to train a mouse lung cell type model based on this combined resource. This model was then applied to assign cell types to count-normalized log transformed data on a per-cell level from mouse lung scRNA-seq samples, using the python scanpy [59] package to normalize total counts per cell to 10,000 and log transform as required by CellTypist.

After initial cell type labelling, further unsupervised clustering of specific cell subtypes was performed, for cells labelled as 'T cells' or 'NK cells' and separately for all antigen presenting cell subtypes, i.e., "Alveolar macrophage", "Dendritic cell", "Monocyte" and "Macrophage". Unsupervised clustering was run using the standard Seurat pipeline which identifies the top 2,000 most variable genes in the data, creates a shared nearest-neighbor (SNN) network of cells, and divides the SNN into discrete clusters using the Louvain algorithm. The resulting clusters were manually annotated by identifying dif-ferentially expressed marker genes for each cluster (using the Seurat FindAllMarkers) function, and linking these marker genes to known cell types, e.g., Cd4+IFNγ+Th1 cells express high levels of Cd3, Cd4 and Ifng).

To quantify changes in cell type proportion over time, total numbers of cells per-sample were calculated and normalized to cells per thousand per sample. Negative-binomial linear models, appropriate for zero-inflated count data, were fit and used to calculate p-values using the R glm.nb function.

Gene expression changes within specific cell types were determined using a pseudobulk approach, where counts from all similarly labelled cells were combined into a single sample x gene count matrix using the Seurat AggregateExpression function. The standard bulk RNAseq analysis package DESeq2 [24] was then used to calculate differential expression foldchanges and p-values for contrasts of interest. Ranked gene lists from the above pseudobulk analysis were used as input for gene-set enrichment analysis using the R fgsea package and mouse MSigDB Hallmark gene sets available in the R msigdbr package. We used the CellChat analysis package (version 1.6.1) to quantify the strength of receptor-ligand communications among cell types in our scRNAseq dataset.

## Flow cytometry

Single cell suspensions were first washed in PBS and then incubated with 50 μl Zombie UV viability dye (BioLegend) for 10 min at room temperature in the dark. Viability dye was immediately quenched by the addition of 100 μl of a surface antibody cocktail diluted in 50% FACS buffer/50% 24G2 Fc block buffer using saturating levels of antibodies. Surface staining was performed for 20 min at 4°C. Then, the cells were washed once with FACS buffer and fixed overnight with the eBioscience Intracellular Fixation and Permeabilization kit (Thermo Fisher). The following day, cells were permeabilized with the provided permeabilization buffer, incubated for 20 min at 4°C with 100 μl of an intracellular antibody cocktail diluted 1:100 in permeabili-zation buffer, and washed with FACS buffer. Cells were analyzed on a BD Symphony A5 cytometer (BD).

## Supporting information

**S1 Table. Per-mouse experimental data.**
(CSV)

**S2 Table. Sp140-/- vs B6 differentially expressed genes per-celltype at day 10.**
(CSV)

**S1 Fig. A. Dotplot showing canonical cell type marker expression per-cell type B. UMAP showing all cell and subsets.**
(PDF)

**S2 Fig. Pre-infection cell proportions in C3H vs B6 mice with and without CoMtb.**
(PDF)

**S3 Fig. Cell type proportions were correlated with Mtb CFU.**
(PDF)

**S4 Fig. Line plots of abundant cell types significantly altered post infection in CoMtb showing A. scRNAseq cells per thousand immune cells B. flow cytometry derived total cell counts and C. flow cytometry derived cells per thousand total cells.**
(PDF)

**S5 Fig. Time dependent DEG counts for SA161 and H37Rv infection (A), and H37Rv GSEA results (B).**
(PDF)

**S6 Fig. UMAPs for B6 and Sp140-/- CoMtb-associated subclusters 10 days post aerosol challenge.**
(PDF)

## Author contributions

**Conceptualization:** Alan H Diercks, Michael Y Gerner, Kevin B Urdahl.

**Data curation:** Fergal J. Duffy, Sara B Cohen, Alan H Diercks.

**Formal analysis:** Fergal J. Duffy, Maxwell L Neal, Courtney R Plumlee.

**Funding acquisition:** Alan H Diercks, Michael Y Gerner, Kevin B Urdahl.

**Investigation:** Fergal J. Duffy, Courtney R Plumlee, Sara B Cohen, Benjamin H Gern, Alan H Diercks, Michael Y Gerner, Kevin B Urdahl, John D Aitchison.

**Methodology:** Fergal J. Duffy.

**Project administration:** Kevin B Urdahl.

**Software:** Fergal J. Duffy.

**Supervision:** Kevin B Urdahl, John D Aitchison.

**Visualization:** Fergal J. Duffy.

**Writing – original draft:** Fergal J. Duffy, Maxwell L Neal, Courtney R Plumlee, Sara B Cohen, Benjamin H Gern, Alan H Diercks, Michael Y Gerner, Kevin B Urdahl, John D Aitchison.

**Writing – review & editing:** Fergal J. Duffy, Maxwell L Neal, Courtney R Plumlee, Sara B Cohen, Benjamin H Gern, Alan H Diercks, Michael Y Gerner, Kevin B Urdahl, John D Aitchison.

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
