## [Decision Letter · Decision Letter 0]

30 Dec 2025

Single-cell profiling reveals that dynamic lung immune responses distinguish protection from susceptibility to tuberculosis

PLOS Pathogens

Dear Dr. Duffy,

Thank you for submitting your manuscript to PLOS Pathogens. After careful consideration, we feel that it has merit but does not fully meet PLOS Pathogens's publication criteria as it currently stands. Therefore, we invite you to submit a revised version of the manuscript that addresses the points raised during the review process.

We look forward to receiving your revised manuscript.

Kind regards,

Anne Jamet

Section Editor

PLOS Pathogens

Anne Jamet

Section Editor

PLOS Pathogens

Editor-in-Chief

PLOS Pathogens

orcid.org/0000-0003-2946-9497

Editor-in-Chief

PLOS Pathogens

orcid.org/0000-0002-7699-2064

**Journal Requirements:**

https://journals.plos.org/plospathogens/s/submission-guidelines#loc-parts-of-a-submission

3) Please insert an Ethics Statement at the beginning of your Methods section, under a subheading 'Ethics Statement'. It must include:

i) The full name(s) of the Institutional Review Board(s) or Ethics Committee(s)

ii) The approval number(s), or a statement that approval was granted by the named board(s).

Potential Copyright Issues:

i) Figure 1A. Please confirm whether you drew the images / clip-art within the figure panels by hand. If you did not draw the images, please provide (a) a link to the source of the images or icons and their license / terms of use; or (b) written permission from the copyright holder to publish the images or icons under our CC BY 4.0 license. Alternatively, you may replace the images with open source alternatives. See these open source resources you may use to replace images / clip-art:

6) Kindly revise your competing statement in the online submission form to align with the journal's style guidelines: 'The authors declare that there are no competing interests.'

**Reviewers' Comments:**

Reviewer's Responses to Questions

**Part I - Summary**

Reviewer #1: The lack of well-established surrogates of protection from Mtb hinders vaccine development. In this submission, Duffy et al. conducted a singe-cell RNAseq analysis of murine lungs following primary Mtb infection or following Mtb challenge in their previously established CoMtb model. Transcriptional responses are measured at three timepoints over 34 days, in response to two Mtb strains, and in both a “susceptible” and a “resistant” mouse strain. Importantly, by studying timepoints soon after infection, the confound of variable Mtb loads is removed. Notable, too, was the appropriate use of B6 sp140-/- mice as comparators with B/6 mice in the CoMtb study.

CoMtb mice exhibited higher expression of IFN-g response genes. Following Mtb challenge, CoMtb mice displayed early macrophage reprogramming as well as increased immune activation and migration signaling. These findings mirror some key features of IV BCG-elicited protection in NHP. Identifying such features associated with protection from Mtb across species helps the important quest to identify robust correlates of protection.

This study addresses an important issue and makes a valuable contribution to the field. This study was well designed and executed; in some cases, transcriptional signatures were verified by flow cytometry. The manuscript is well-written and the figures are appropriate and well laid-out. There are just a few concerns that should be addressed.

Reviewer #2: This is a comprehensive single cell analysis of the lungs of Mtb aerosol infected mice at early time points with a view to identifying correlates of protection. The experimental work is performed well with sufficient numbers of animals, a reproducible infection model and suitable hypotheses under test.

The approach is not novel - there are several studies already published in this arena - however there are elements that are important to the field of TB immunity. The comparison between mouse strains and genetically deficient strains as well as the comparison between naïve mice and mice with concomitant immunity are suitable comparisons and rationally based.

The strengths lie in the scope of the work and the in depth analysis - the data sets will be of use to the field as a whole for both model development and extension into human TB interventions.

The weaknesses lie in the presentation and the failure to deliver causal connections between the observations and the outcomes. The mouse model is useful because it allows direct mechanistic assessment of the immune response to infection over time, dose and route; these definitive outcomes can then be integrated into models wherein these causal connections cannot be made.

1. The presentation of the paper indicates rushed presentation. There are numerous grammatical errors and the text of the results section has the feel of a laboratory group presentation. A thorough re-edit of the results section with removal of the technical discussions - i.e. sort out how to account for the differences in baseline between the mouse strains and naïve/concomitant groups rather than stating it as problem for analysis.

2. Some of the data is discussed as being in the figures but is not shown - also the figures are not referenced correctly i.e. Figure 1 C does not show day 10 counts for all conditions and it is referred to as Fig 1B in places. Figure 3 has 3C before 3B in the figure layout. These are just things that should have been addressed in the editing process by the senior author.

3. Overall the presentation of the results suffers from having too many variables lumped together in presentation. This makes it exceptionally difficult to present the outcomes clearly, and importantly, the quality and impact of the data sets is lost to the reader. Pick a subject, explain it from start to finish and then move to the next subject. I would address the differences between the primary response of the strains first B6 to C3H then sp140, then move to the impact of concomitant immunity. Trying to bring all these together to generate a single (or simple) correlate of protection would have worked if there was one - but there isn't one common theme, and therefore just a straightforward description of the data is more helpful to the field and to our understanding of TB immunity.

4. Most of the outcomes noted are as expected but there are some novel observations such as the transcriptional differences in the myeloid cells. It would be important however to make some causal connection between any one of the observations reported and the altered immunity. i.e. removal of a cell type, disruption of a communication channel etc.

5. The data sets are largely transcriptional in nature - it would be very helpful for the figure legends to be explicit about the technical elements of the data shown - what is flow data, what is transcriptional data, what are the n numbers? how many experiments were performed and how much of the data is shown

6. One technical issue is that of the potential differences in the inoculum between the SA161 and the H37Rv used for aerosol. Can the group be certain the presentation of bacteria to the innate immune response is the same between the two during the first 10 days - should be addressed in the text just to show it has been considered.

Overall good source of data but presentation leaves the field more complex rather than clarified.

**Part II – Major Issues: Key Experiments Required for Acceptance**

Please use this section to detail the key new experiments or modifications of existing experiments that should be absolutely required to validate study conclusions.required to validate study conclusions.required to validate study conclusions.required to validate study conclusions.

Reviewer #1: None

Reviewer #2: The data sets are of interest but there is no causal connection made between the observations and the outcomes. Is there any data set that could be supplied that would show the importance of any one of the new pathways identified to protection?

While a computational tool could be used to make quasi-causal connections, this would not be the best option.

**Part III – Minor Issues: Editorial and Data Presentation Modifications**

Reviewer #1: Fig 1C shows “Mtb lung CFU post infection over time, stratified by mouse strain, Mtb strain, and CoMtb status”. However, the figure appears to differentiate only Mtb strain and mouse status (Primary v CoMtb). Which dots represent B6 and which represent C3H? Why is there no CFU data at 10 days following infection with H37Rv, which is important to support the contention that bacterial burden is similar at this time point.

Line 43: The term “symptomatic” implies self-reported feelings of illness. The more correct term is “signs of disease”.

Mice were aerogenically infected with 50-100 CFU, which is often used in the field, but is supraphysiologic.

Reviewer #2: This is not really a minor issue but the presentation of the data and the results section compromises the clarity and impact of the data. However as this can be addressed by effective senior editing then it is a minor issue. Pleaser ensure the data is presented rationally in sequence with single variables addressed clearly. This can be followed by integration of the data - and even some novel computational analysis to bring the observations together would be helpful.

PLOS authors have the option to publish the peer review history of their article (what does this mean?). If published, this will include your full peer review and any attached files.). If published, this will include your full peer review and any attached files.). If published, this will include your full peer review and any attached files.). If published, this will include your full peer review and any attached files.

...

Reviewer #1: No

Reviewer #2: No

**Figure resubmission:**While revising your submission, we strongly recommend that you use PLOS’s NAAS tool (https://ngplosjournals.pagemajik.ai/artanalysis) to test your figure files. NAAS can convert your figure files to the TIFF file type and meet basic requirements (such as print size, resolution), or provide you with a report on issues that do not meet our requirements and that NAAS cannot fix.

**Reproducibility:**



---

## [Decision Letter · Decision Letter 1]

7 Apr 2026

Dear Dr. Duffy,

We are pleased to inform you that your manuscript 'Single-cell profiling reveals that dynamic lung immune responses distinguish protection from susceptibility to tuberculosis' has been provisionally accepted for publication in PLOS Pathogens.

Best regards,

Helena Ingrid Boshoff

Section Editor

PLOS Pathogens

Anne Jamet

Section Editor

PLOS Pathogens

Sumita Bhaduri-McIntosh

Editor-in-Chief

PLOS Pathogens

orcid.org/0000-0003-2946-9497

Michael Malim

Editor-in-Chief

PLOS Pathogens

orcid.org/0000-0002-7699-2064

Reviewer Comments (if any, and for reference):

Reviewer's Responses to Questions

**Part I - Summary**

Reviewer #2: (No Response)

**Part II – Major Issues: Key Experiments Required for Acceptance**

Please use this section to detail the key new experiments or modifications of existing experiments that should be absolutely required to validate study conclusions.required to validate study conclusions.required to validate study conclusions.required to validate study conclusions.

Reviewer #2: (No Response)

**Part III – Minor Issues: Editorial and Data Presentation Modifications**

Reviewer #2: (No Response)

PLOS authors have the option to publish the peer review history of their article (what does this mean?). If published, this will include your full peer review and any attached files.). If published, this will include your full peer review and any attached files.). If published, this will include your full peer review and any attached files.). If published, this will include your full peer review and any attached files.

...

Reviewer #2: No

---

## [Editor Report · Acceptance letter]

Dear Dr. Duffy,

We are delighted to inform you that your manuscript, "Single-cell profiling reveals that dynamic lung immune responses distinguish protection from susceptibility to tuberculosis," has been formally accepted for publication in PLOS Pathogens.

Best regards,

Sumita Bhaduri-McIntosh

Editor-in-Chief

PLOS Pathogens

orcid.org/0000-0003-2946-9497

Michael Malim

Editor-in-Chief

PLOS Pathogens

orcid.org/0000-0002-7699-2064